# Alterations in Cerebellar Microtubule Cytoskeletal Network in a ValproicAcid-Induced Rat Model of Autism Spectrum Disorders

**DOI:** 10.3390/biomedicines10123031

**Published:** 2022-11-24

**Authors:** Magdalena Gąssowska-Dobrowolska, Agnieszka Kolasa, David Q. Beversdorf, Agata Adamczyk

**Affiliations:** 1Department of Cellular Signalling, Mossakowski Medical Research Institute, Polish Academy of Sciences, Pawińskiego 5, 02-106 Warsaw, Poland; 2Department of Histology and Embryology, Pomeranian Medical University, Powstańców Wlkp. 72, 70-111 Szczecin, Poland; 3Department of Radiology, Neurology, and Psychological Sciences, University of Missouri, DC069.10, One Hospital Drive, Columbia, MO 65279, USA

**Keywords:** α/β-tubulin, MAP-Tau, MAP1B, MAP2, MAP6 (STOP), αII-spectrin, Tau-kinases, valproic acid (VPA), autism spectrum disorders (ASD), cerebellum

## Abstract

Autism spectrum disorders (ASD) are neurodevelopmental diseases characterised by deficits in social communication, restricted interests, and repetitive behaviours. The growing body of evidence points to a role for cerebellar changes in ASD pathology. Some of the findings suggest that not only motor problems but also social deficits, repetitive behaviours, and mental inflexibility associated with ASD are connected with damage to the cerebellum. However, the understanding of this brain structure’s functions in ASD pathology needs future investigations. Therefore, in this study, we generated a rodent model of ASD through a single prenatal administration of valproic acid (VPA) into pregnant rats, followed by cerebellar morphological studies of the offspring, focusing on the alterations of key cytoskeletal elements. The expression (Western blot) of α/β-tubulin and the major neuronal MT-associated proteins (MAP) such as MAP-Tau and MAP1B, MAP2, MAP6 (STOP) along with actin-crosslinking αII-spectrin and neurofilament light polypeptide (NF-L) was investigated. We found that maternal exposure to VPA induces a significant decrease in the protein levels of α/β-tubulin, MAP-Tau, MAP1B, MAP2, and αII-spectrin. Moreover, excessive MAP-Tau phosphorylation at (Ser396) along with key Tau-kinases activation was indicated. Immunohistochemical staining showed chromatolysis in the cerebellum of autistic-like rats and loss of Purkinje cells shedding light on one of the possible molecular mechanisms underpinning neuroplasticity alterations in the ASD brain.

## 1. Introduction

Autism spectrum disorders (ASD) are a heterogeneous group of neurodevelopmental conditions diagnosed in more than 1% of children with onset very early in life and a complex, multifactorial aetiology. ASD are characterised by impaired social interaction and deficits in communication skills, together with the presence of restricted and/or stereotyped patterns of behaviours, activities, or interests with a high incidence of abnormalities in sensory reactivity [1,2]. Although ASD share some common behavioural symptoms, their pathogenesis has not yet been completely elucidated, and their underlying causes are very diverse. Risk factors that increase the likelihood of ASD in a child include genetic and environmental factors, as well as the interactions between them [3,4]. Exposure to stress, toxins, drugs, environmental pollutants, or maternal infections during the early phase of embryonic life and immediately after birth are involved in the aetiology of ASD via activation of the maternal immune system, leading to atypical foetal brain development and a range of congenital malformations [5,6].

The pathogenesis of ASD is not yet fully understood, but increased research efforts have provided evidence that abnormal brain development (such as synaptic and myelin dysfunction, anomalies in neuronal cell architecture, and compromised neuronal connectivity) may underlie the aetiology of ASD [7,8,9,10,11]. Albeit the genetic screening has identified hundreds of mutations and other genetic abnormalities associated with ASD (https://gene.sfari.org/database/human-gene/; http://autismkb.cbi.pku.edu.cn/index.php (accessed on 12 October 2022)), a growing body of evidence suggests that many of these alterations are linked with genes encoding microtubule (MT)-associated proteins, cytoskeletal elements and synaptic molecules that affect various aspects of neuronal communication [12,13,14,15,16,17,18,19,20,21]. This genetic evidence—together with significant morphological abnormalities within axons, dendrites, dendritic spines, and in the organisation of the neural networks in ASD subjects—suggests that different cytoskeletal filaments could be compromised in ASD [22,23,24,25,26,27,28]. The proper formation and development of the nervous system, as well as suitable brain connectivity, involve extremely complex processes governed by the communication and careful coordination of the neuronal cytoskeleton, comprising a three-dimensional lattice of the three main families of filaments: MTs, actin-based microfilaments (AF), and neurofilaments (NF) [17].

MTs are one of the major structural components of the cytoskeleton, present in all eukaryotic cell types that are comprised of α- and β-tubulin heterodimers (MT building blocks) [14,29]. Extremely dynamic MTs are crucial for the development and stabilisation of axonal and dendritic processes, neurogenesis, synaptogenesis, as well as neurotransmission. The elaborate MTs network is essential for neuronal growth, morphology, migration, and cell polarity. In mature neurons, the MT networks also serve as tracks for long-distance intracellular cargo trafficking along axons and dendrites to establish appropriate signal transduction and neural connectivity [11,14,17,30,31]. The structure of MTs and their extremely dynamic properties are heavily dependent on the cycles of assembly (growing state/polymerisation), and disassembly (shrinking state/depolymerisation) and is adjusted by post-translational α/β-tubulin modifications as well as by interaction with different MT-binding/MT-associated proteins (MAP) [17,32]. MAPs are divided into several categories, such as MT motors, the MT-severing protein katanin, and structural MAPs, which include MAP-Tau, MAP1, MAP2, and MAP6 protein, also known as STOP (stable-tubule-only polypeptide) [11,16,33,34].

One of the key regulators of MT dynamics, mainly found in the axonal compartment, is the structural MT-associated (MAP) Tau protein [35]. The pivotal role of Tau is to protect MTs against depolymerisation by reducing the dissociation of α/β-tubulin at both MT ends [36]. In this way, Tau is of particular importance in the stabilisation of axonal MTs, thus determining proper axonal transport, neurite outgrowth, and synapse formation [37,38,39]. The affinity of Tau for the MTs depends on its phosphorylation status [40,41]. In a normal brain, the low phosphorylation of Tau enables its interplay with α/β-tubulin in order to trigger its assembly into MT and help stabilise its structure [27,39]. Hyperphosphorylation of Tau decreases its MT-binding capacity and disrupts MT stability [41,42,43,44,45]. Abnormal phosphorylated Tau exhibits prion-like activity, sequestrating not only normal Tau but also the other neuronal MAPs, such as MAP1 and MAP2, and destroying pre-assembled MTs, leading to progressive degeneration of the affected neurons [46,47].

The classical MAPs that bind along the MT lattice are MAP1 family proteins. MAP1B is expressed in axons, dendrites, and growth cones, where it regulates axonal guidance and elongation and neuronal migration, as well as participates in the regulation of the structure and physiology of dendritic spines in glutamatergic synapses. MAP1B is a known modulator of MT dynamics and stability. Some studies suggest that MAP1B mediates MT stabilisation specifically by reducing depolymerisation rates [33,48,49,50]. The next protein from the structural MAPs category, expressed mainly in neuronal dendrites, is MAP2, which, similarly to Tau, can interact with MTs through its α/β-tubulin-binding domain and regulate neurite outgrowth, MT dynamics, and organelle transport in axons and dendrites [34,39]. The next MAP–MAP6 (STOP) is a calmodulin-regulated protein responsible for the high degree of stabilisation of MT by bridging the binding between adjacent MTs, and the establishment of neuronal architecture and synaptic plasticity [29]. The STOP protein is responsible for regulatory functions in neuronal cells, including the regulation of both actin and MT dynamics in axons and dendrites, as well as in dendritic spines and synaptic protein complexes [51].

In addition to the MAPs, the neuronal cytoskeleton is formed by the two other main families of filaments: actin filaments (F-actin) and neurofilaments (NF) [52]. F-actin is one of the main components of the cytoskeleton concentrated in dendritic spines that governs spine morphogenesis and plasticity, neurite outgrowth, and axonal guidance [53,54]. In turn, NF, and especially neurofilament light polypeptide (NF-L), are key skeletal proteins linked directly to several synaptic proteins, thereby regulating synaptogenesis, neuronal structure, and neurotransmission [52].

MTs, actin filaments, and neurofilaments interact to provide a proper formation of the nervous system during embryonic development and to ensure its function in adulthood. There is sufficient evidence to suggest that destabilisation of the MTs system and the neuronal cytoskeleton could lead to synaptic malfunction and eventually to the pathogenesis of neurodevelopmental disorders such as ASD [11,14,17,29]. Abnormalities in synaptic function could mediate the formation of autistic-like behaviour [55]. Changes in various proteins associated with the MT cytoskeleton have already been revealed in many neurodevelopmental disorders as well as in individuals with ASD and in different experimental models of ASD [11,14,17,19,24,56], but studies on the involvement of the fundamental MT building block -α/β-tubulin and MAP-Tau in the pathomechanisms of ASD are still relatively sparse. To date, alterations in both the level of Tau as well as its excessive phosphorylation have been the subject of intense research in neurodegenerative diseases, in particular in Alzheimer’s disease (AD). Only our latest research indicated that maternal exposure to VPA induces Tau accumulation in the hippocampus and cerebral cortex of adolescent rat offspring, together with excessive Tau phosphorylation in a brain-region-dependent manner [57].

Since cerebellum controls not only motor skills but also supports cognitive and language functions, and as the latest research suggests cerebellar pathology in ASD [58,59,60,61], in the current study, we investigated alterations in cerebellar MT cytoskeletal network in a VPA-induced rat model of ASD.

We found that prenatal VPA exposure led to a decrease in the levels of a wide spectrum of structural MT-associated proteins, including neuronal α/β-tubulin, MAP-Tau, MAP1B, MAP2A/B, MAP2C/D, and actin-crosslinking protein-αII-spectrin. Moreover, it has been shown excessive Tau phosphorylation at (Ser396) together with activation of CDK5, AMPK, and p70S6 kinases. Along with molecular changes, histopathological abnormalities were observed in the cerebellar neurons (chromatolysis) and a significant loss of Purkinje cells in VPA offspring. Altogether, the observed deregulation of key cytoskeletal protein homeostasis and the pathological alterations in the morphology of neurons imply the dysfunction and, thus, destabilisation of the cytoskeletal network, which may lead to a disturbance of synaptic connectivity, contributing to dysfunctional behaviours.

## 2. Materials and Methods

### 2.1. Ethical Statement

All experiments conducted with animals were approved by the Local Ethics Committee for Animal Experimentation in Warsaw, Poland (reference number 4/2014, 60/2015, 64/2015, 361/2017 WAW2/083/2018 and WAW2/148/2018), and were carried out in accordance with the EC Council Directive of 24 November 1986 (86/609/EEC), following the ARRIVE guidelines, and guidelines published in the NIH Guide for the Care and Use of Laboratory Animals, and the principles presented in the “Guidelines for the Use of Animals in Neuroscience Research” by the Society for Neuroscience. Efforts were made to minimise animal suffering and to reduce the number of animals used. All manipulations were performed gently and quickly to avoid stress-induced alterations.

### 2.2. Animals-In Vivo Model of ASD

Pregnant Wistar rats between 12 and 15 weeks of age and weighing 210–250 g were supplied by the Animal House of the Mossakowski Medical Research Centre, Polish Academy of Sciences (Warsaw, Poland), which operates breeding of small rodents with the SPF standard. The animals were maintained under controlled conditions of temperature and humidity with a 12-h light/dark cycle. Procedures involving animals were carried out in strict accordance with international standards of animal care guidelines, and every effort was made to minimise suffering and the number of animals used. According to the study by Schneider and Przewlocki [62], the ASD model was induced by a single intraperitoneal injection (i.p.) of VPA at a dose of 450 mg/kg of body weight on gestational day 12.5 (experimental group). Pregnant females from control groups received a single i.p. of solvent (sterile 0.9% NaCl). All pregnant dams were allowed free access to food and water and were kept in a room with a controlled temperature under an LD 12/12 regime to give birth. All dams were allowed to give birth and nurture offspring under normal conditions. The day of birth was considered a postnatal day (PND) 1. On PND 7, each litter was equalised (random selection), and the number of pups was limited to 10 (both male and female). Offspring (males and females) stayed with their mothers and were fed by them until PND 22–23, and then rat pups were separated and kept in groups of 3 or 4 in open polycarbonate cages in an enriched environment. To avoid interference from the hormonal disturbances/changes, only males were selected for further experimental procedures. Adolescent males were sacrificed at PND 58–59 by decapitation; their brains were quickly removed and dissected into the cerebellum and then placed in liquid nitrogen. The samples were stored at −80 °C for further Western blot analysis. To avoid the litter effect, animals from three different litters in each experimental group (random selection) were used for the experiments.

### 2.3. Immunochemical Determination of Protein Levels (Western Blot Analysis)

Immunochemical analysis of protein level and phosphorylation status was performed by the Western blotting method in standard conditions. Tissue samples were homogenised, mixed with Laemmli buffer, and denatured at 95 °C for 5 min. After standard SDS-PAGE separation, the proteins were “wet”-transferred to nitrocellulose membranes in standard conditions and used for immunochemical analysis with specific antibodies, followed by chemiluminescent detection. The membranes were washed for 5 min in TBST (Tris-buffered saline with Tween 20 buffer: 100 mM Tris, 140 mM NaCl and 0.1% Tween 20, pH 7.6), and non-specific binding was blocked for 1 h at room temperature (RT) with 2% or 0.5% BSA in TBST or with 5% non-fat milk solution in TBST. Membranes were probed with the following primary antibodies: α/β-tubulin (1:1000), Tau (1:500), pTau(Ser396) (1:250), pTau(Ser199/202) (1:1000), pTau(Ser416) (1:1000), pGSK-3β(Ser9) (1:250), pGSK-3β(Ser389) (1:500), pGSK-3β(Tyr216) (1:250), GSK-3β (1:1000), pp44/pp42MAPK(Thr202/Tyr204) (1:1000), p44/p42MAPK (1:1000), p35/p25 (1:1000), p-AMPK(Thr172) (1:500), AMPK (1:500), MAP1B (1:500), MAP2 (1:1000), p-MAP2(Ser136) (1:1000), MAP6 (STOP) (1:250), NF-L (1:125), αII-spectrin (1:1000) (Appendix A in Appendix A). The membranes were washed three times in TBST, incubated for 60 min at RT with appropriate secondary antibodies (1:8000 anti-rabbit or 1:4000 anti-mouse IgG), and washed again 3 × in TBST. Antibodies were detected using chemiluminescent reaction and ECL reagent (Amersham Biosciences, Bath, UK) under standard conditions. After each protein detection, the membranes were stripped (25 mM Glycine-HCl, 1% (*w/v*) SDS, pH 2; 30 min at room temperature) and re-probed. First, phosphorylated protein was immunodetected, then the total level of analysed protein, and finally, glyceraldehyde 3-phosphate dehydrogenase (GAPDH) or vinculin as a loading control. GAPDH (1:50,000) was used as a loading control for low-molecular-weight proteins; vinculin (1:1000) was used as a loading control for analyses of high-molecular-weight proteins. In all experiments, densitometry analysis of immunoblots was performed using normalisation to immunoreactivity of GAPDH or vinculin. Densitometric analysis and size-marker-based verification were performed with TotalLab software.

### 2.4. Immunohistochemistry Analysis

After decapitation, the dissected brains were fixed in formalin. Then, the brains were washed with absolute ethanol (3 times within 3 h), absolute ethanol with xylene (1:1) (twice within 1 h), and xylene (3 times within 20 min). Then, following 3 h of saturation of the tissues with liquid paraffin, the samples were embedded in paraffin blocks. Using a microtome (Microm HM340E), 3–5 µm serial sections were cut and placed on polysine histological slides (Thermo Scientific, J2800AMNZ, Waltham, MA, USA). The sections of the brains were deparaffinised in xylene and rehydrated in decreasing concentrations of ethanol and then used for immunohistochemical (IHC) staining. In order to expose the epitopes, the sections were boiled twice in a microwave oven (700 W for 4 and 3 min) in 10 nM citrate buffer (pH 6.0). Once cooled and washed with PBS, the endogenous peroxidase was blocked by a 3% solution of perhydrol in methanol, and then the slides were incubated over the night at 4 °C with primary antibodies. To visualise the antigen-antibody complex, a Dako LSAB+System-HRP was used (DakoCytomation, K0679, Glostrup, Denmark), based on the reaction of avidin-biotin-horseradish peroxidase with DAB as a chromogen, according to the included staining procedure instruction. Sections were washed in distilled water and counterstained with hematoxylin. For negative control, specimens were processed in the absence of primary antibodies. Positive staining was defined microscopically (Leica DM5000B, Wetzlar, Germany) by visual identification of brown pigmentation. Primary antibody list: Tau (Santa Cruz Biotechnology, Dallas, TX, USA, sc-32274, final dilution 1:500), pTau(Ser416) (Cell Signalling, Danvers, MA, USA, 15013, 1:200), pTau(Ser396) (Cell Signalling, Danvers, MA, USA, 9632, 1:200), pTau(Ser199/202) (Sigma Aldrich, Saint Louis, MO, USA, T6819, 1:1000), α/β-tubulin (Cell Signalling, Danvers, MA, USA, 2148S, 1:100). The paraffin slides (3 μm) stained with hematoxylin-eosin (H&E) underwent a general histological examination. The samples were independently examined by two experienced pathologists.

Additionally, the Purkinje cell numbers were counted. From each rat (*n* = 5 per control and *n* = 5 per VPA group), three photomicrographs in the representative places of the cerebellum were taken, and all Purkinje cells visible in the field of view were counted and subjected to statistical analysis.

### 2.5. Statistical Analysis

The results are expressed as mean values ± S.E.M. In all the analyses, each data point is from a separate animal. The normality and equality of the group variances were tested using a Shapiro–Wilk test. Differences between the means were analysed using an unpaired Student’s t-test for data with normal distributions; or a non-parametrical Mann–Whitney U test for data with non-normal distribution. Differences were considered significant at *p* < 0.05. The statistical analyses were performed using Graph-Pad Prism version 8.0 (Graph Pad Software, San Diego, CA, USA).

## 3. Results

### 3.1. Prenatal Exposure to VPA Induces Changes in the Level of α/β-Tubulin and MAP-Tau in the Cerebellum of Adolescent Rat Offspring

To evaluate the expression of key proteins responsible for the regulation of MT assembly and stability in the cerebellum of autistic-like rats, we analysed the protein level of α/β-tubulin and MAP-Tau. Using the Western blot technique, we revealed that prenatal exposure to VPA significantly decreased the level of both α/β-tubulin by about 19% (*p* = 0.0034) and total Tau protein by about 38% (*p* = 0.0005) in the cerebellum of VPA offspring, compared to the control group (Figure 1A,B). The Western blot results were confirmed by immunohistochemical analysis. In the cerebellum of control rats, the immunoexpression of α/β-tubulin was significantly more intense than in VPA-exposed rats (Figure 2) and was visible in the neurons of all three layers (molecular layer (ML); the Purkinje cell layer (PL); and the granular layer (GL)) of the cerebellum (blue, red and yellow arrows, respectively). After exposure to VPA, only a few cells of the granular layer showed a very weak expression of α/β-tubulin (Figure 2; yellow arrows), and Purkinje cells were immunonegative (Figure 2; white arrows). Similarly, the immunohistochemical analysis of Tau protein revealed a little bit lower expression of this protein in VPA-exposed animals compared to control rats (Figure 2).

### 3.2. Prenatal Exposure to VPA Leads to Alterations in the Phosphorylation Status of MAP-Tau Protein in the Cerebellum of Adolescent Rat Offspring

Because both the function and affinity of Tau for the MT depend on its phosphorylation state, in the next step, we measured the phosphorylation status of Tau at the (Ser396), (Ser199/202), and (Ser416) epitopes. Prenatal exposure to VPA significantly (*p* = 0.0082) increased the level of pTau(Ser396) by about 40% compared to the control group (Figure 3A) without effect on (Ser199/202) phosphorylation (Figure 3B). However, the phosphorylation status at (Ser416) was significantly decreased (by about 28%, *p* = 0.0128) (Figure 3C). The Western blot analysis was confirmed by the immunohistochemical investigation. The immunoexpression of pTau, phosphorylated at (Ser396), was apparently higher in the neurons of all three layers of the cerebellum in VPA-exposed rats than in control (Figure 4). In particular, the neuropil of the granular layer was characterised by the high immunoexpression of this protein (Figure 4). In turn, the expression of pTau(Ser199/202) in the neurons of the cerebellum in VPA-exposed rats was quite similar to the control group (Figure 4). However, we observed that in the VPA offspring rats, Purkinje cells were immunopositive (red arrow) in contrast to immunonegative neurons (white arrows) in control animals (Figure 4). Moreover, the intensity of pTau(Ser416) expression in the neurons of the cerebellum in the VPA-exposed group was a little bit lower than in control rats (Figure 4), despite the fact that Purkinje cells expressed pTau(Ser416) in VPA-treated rats (red arrows) (Figure 4) in contrast to immunonegative neurons in control rats (white arrows) (Figure 4).

### 3.3. Prenatal Exposure to VPA Leads to the Deregulation of Tau Kinases Activity in the Cerebellum of Adolescent Rat Offspring

To evaluate the involvement of the major Tau kinases in VPA-induced Tau phosphorylation, the activity of CDK5, GSK-3β, ERK1/2, AMPK, and p70S6K was analysed.

The best-known mechanism of detrimental overactivation of CDK5 is the calcium-induced calpain-dependent proteolytic cleavage of p35–p25 protein. Therefore, here we analysed the p35 level and its cleavage to p25. As presented in Figure 5A,B, exposure to VPA during foetal life significantly (*p* = 0.0488) reduced the level of p35 protein by about 9%, associated with increased the formation of truncated p25 protein (by about 41%, *p* = 0.0022). The elevated p25/p35 ratio (by about 1.4 times, *p* = 0.0048) (Figure 5C) suggests that VPA activates calpain-dependent cleavage of p35. The stimulatory effect of VPA on the calpain-dependent activation of the CDK5/p25 complex was confirmed by analysing the level of 145 kDa spectrin breakdown product (SBDP) protein, which is formed as a result of the calpain-catalysed breakdown of actin-crosslinking protein, αII-spectrin. As shown in Figure 5D, prenatal exposure to VPA induced a significant increase in the level of SBDP (by about 160%, *p* = 0.0236). Thus, prenatal exposure to VPA significantly stimulated CDK5 activity in the cerebellum of VPA offspring, suggesting the involvement of CDK5 in VPA-induced Tau hyperphosphorylation.

Since VPA exposure had no effect on GSK-3β as well as on ERK1/2 activity, we excluded the involvement of these enzymes in VPA-evoked Tau phosphorylation (Appendix A in Appendix A).

In the next step, we examined the activity of AMPK as a kinase that phosphorylates Tau on a number of sites, including the (Ser396) epitope, altering the MT-binding of Tau. Because the α-subunit of AMPK is the catalytic subunit, and the phosphorylation of the (Thr172) in this subunit is a hallmark of AMPK activation, we analysed the level of phosphorylated AMPK at (Thr172) together with the level of total AMPK. Our study indicated a significant (*p* = 0.0135) increase in the immunoreactivity of p-AMPK, phosphorylated on (Thr172), by about 29% (Figure 6A), whereas the protein level of AMPK was unchanged (Figure 6B) in the cerebellum of VPA-exposed rats, compared to control. The data indicate stimulation of AMPK activity in the cerebellum of VPA offspring, suggesting the potential involvement of this kinase in the VPA-induced excessive phosphorylation of Tau.

We further investigated p70S6K as the next kinase potentially involved in Tau hyperphosphorylation. The activity of p70S6K was determined by measurement of the phosphorylation status at (Ser371). Additionally, the protein level of p70S6K was analysed. As presented in Figure 7A, prenatal exposure to VPA led to a significant (*p* = 0.0386) increase in the level of phospho-p70S6K(Ser371) by about 25%, compared to the control group. p70S6K is one of the downstream effectors of mTOR, and so to confirm the VPA-dependent activation of p70S6K, we also analysed the mTOR activity by measuring the level of phosphorylated mTOR at (Ser2448) as an established marker of mTORC1 activation. Moreover, the level of total mTOR and the p-mTOR(Ser2448)/mTOR ratio were determined. As shown in Figure 7B and C, prenatal exposure to VPA evoked a significant (*p* = 0.0153) increase in the level of p-mTOR(Ser2448) by about 134% without changes in the level of total mTOR. Moreover, the ratio of p-mTOR(Ser2448)/mTOR was markedly raised by about 70% (*p* = 0.0101) in the cerebellum of VPA adolescent rat offspring (Figure 7D).

### 3.4. Prenatal Exposure to VPA Induces Changes in the Level of the Other Structural MAPs in the Cerebellum of Adolescent Rat Offspring

Since the MT defects and deregulation of the MT cytoskeletal network are linked to several neurological conditions, including ASD, we investigated whether prenatal exposure to VPA affects the other structural MAP proteins. In particular, we examined the protein level of MAP1B, MAP2, and MAP6 (STOP). As shown in Figure 8A, the immunoreactivity of the MAP1B heavy chain (271 kDa) was markedly decreased by about 71% (*p* = 0.0002) in the cerebellum of VPA adolescent offspring. Similarly, the prenatal exposure to VPA evoked a significant (*p* < 0.0001) decrease in the level of both high-molecular-weight MAP2 (which includes MAP2A and MAP2B, with molecular masses of 280 and 270 kDa, respectively) by about 89% and low-molecular-weight MAP2 (which includes MAP2C and MAP2D, which molecular masses of 70 and 75 kDa, respectively) by about 40% (*p* < 0.0001), compared to the respective control groups (Figure 8B,C). The function of the MAP2 protein depends on its phosphorylation state. Similar to Tau protein, abnormal hyperphosphorylation of MAP2 decreases its binding affinity to MT (association with α/β-tubulin) and disrupts MT stability. Therefore, we analysed the phosphorylation status of MAP2A/B and MAP2C/D at a specific (Ser136) residue. Using Western blot analysis, we found a significant (*p* < 0.0001) decrease in the phosphorylation of both *p*-MAP2A/B(Ser136) by about 71% as well as p-MAP2C/D(Ser136) by about 51% (*p* = 0.0136) (Figure 8D,E). In turn, no difference was found in the amount of MAP6 (STOP) protein (Figure 8F).

### 3.5. Prenatal Exposure to VPA Induces Changes in the Level of the Other Major Cytoskeletal Proteins in the Cerebellum of Adolescent Rat Offspring

Besides MT, the elements which guarantee a proper formation of the nervous system during embryonic development and assure its function in adulthood are neurofilaments and actin filaments and their mutual interactions. One of the major neurofilaments in the axons, forming the core of the NF bundle, as well as the critical scaffolding molecule of the neurite extensions, which regulates the radial diameter of axons and the overall shape of the neurocytoskeleton, is the NF-L protein. The major cytoskeletal protein of most cells is actin, and an actin-crosslinking protein is αII-spectrin—one of the critical components of the actin cytoskeleton. Therefore, in the next step, we examined the effect of VPA on the level of these proteins. We revealed that exposure to VPA during embryonic development had no effect on the protein level of NF-L (Figure 9A). In turn, the immunoreactivity examination of αII-spectrin revealed a significant depletion in the level of this protein by about 42% (*p* = 0.0013) in the cerebellum of VPA offspring, compared to control (Figure 9B). This considerable decrease in αII-spectrin level was accompanied by an elevated level of SBDP (Figure 5D), which is generated as a consequence of the calpain-catalysed breakdown of αII-spectrin. In addition, the ratio of SBDP/αII-spectrin was significantly increased in the cerebellum of VPA rats (this ratio was about 2 times greater in VPA offspring than in control, *p* < 0.0001) (Figure 9C).

### 3.6. Prenatal Exposure to VPA Induces Loss of Purkinje Cells along with Histopathological Changes in the Neurons

Disturbances in the levels and phosphorylation of the cytoskeletal proteins observed in the cerebellum of adolescent rat offspring exposed prenatally to VPA encouraged us to investigate the possible effects of this compound on the structure of nerve cells. Moreover, we analysed the effect of VPA exposure on the number of Purkinje cells. Our study revealed a significant (*p* = 0.0116) loss of Purkinje cells (by about 28%) in the cerebellum of VPA-exposed adolescent offspring compared to control (Figure 10A,B).

The histological analysis of neurons (Purkinje cells) in the cerebellum of control rats indicated no structural changes in nerve cells. As presented in Figure 11A (blue arrows), neurons had a normal appearance and structure with basophilic cytoplasm in perikaryons, well-defined Nissl bodies, as well as the nucleus in central position. In turn, the prenatal administration of VPA induced pathological changes in the structure of Purkinje cells in the examined brain region. These neurons exhibited the pathological phenotype (chromatolysis), which was manifested as eosinophilic cytoplasm in perikaryon, acidophilic degradation of Nissl bodies, and presence of pyknotic nucleus located eccentrically near the cell membrane (Figure 11B,C, red arrows). These kinds of cells were also observed in control rats (Figure 11A, red arrow), but they were very rare in comparison to VPA-treated rats. Moreover, near the Purkinje cell layer, there was an area infiltrated by mononuclear small, dark staining cells resembling lymphocytes (cells with dark staining nucleus and light halo envelope; Figure 11B, black circles/ellipsoid), and this area showed incorrect organisation of neuropil (Figure 11C, green frame). All these observations suggest that the cytoskeletal network was negatively affected by VPA.

## 4. Discussion

Several clinical and animal studies have revealed the possible involvement of the cerebellum in ASD pathology [58,59,60,61]. Individuals with ASD showed a significant cerebellar pathology, including a reduction of Purkinje cells [58,63,64,65]. In line with the results of, e.g., Wang et al., Han-Sam et al., and Shona et al. [59,65,66], our study supports and extends evidence for the neurotoxic effect of prenatal VPA exposure on the cerebellar cortex, especially on Purkinje cells, whose number was significantly reduced by about 28%.

The current study is the first investigation where we provide evidence that in the cerebellum of an ‘autistic’ rat, after embryological exposure to VPA comes defects in MT assembly and properties of the MT cytoskeleton. We revealed a significant decrease in the level of α/β-tubulin as well as major MAP proteins such as Tau, MAP1B, MAP2A/B, and MAP2C/D, as well as cytoskeletal actin-crosslinking αII-spectrin. Additionally, we demonstrated Tau protein hyperphosphorylation at (Ser396) that was accompanied by the deregulation of activity of some Tau kinases (CDK5, AMPK, and p70S6K). Dyshomeostasis of MAP proteins and histopathological changes in Purkinje neurons (chromatolysis) suggest that prenatal exposure to VPA may disrupt the MT network and all cytoskeletal integrity, resulting in the impairment of synaptic structure, function, and plasticity that lead to behavioural changes and neurodevelopmental deficits.

Cytoskeletal elements, such as the MT network, are indispensable for correct brain connectivity and the proper functioning of synapses and neurons by regulating morphogenesis, synaptogenesis, neurite growth, neuronal branching, polarisation, migration, and myelination [17,30]. In cooperation with several classes of MAP proteins, MTs regulate the long-distance intracellular trafficking of signalling molecules and trophic factors along axons and dendrites [32]. MTs are also essential in the growth and guidance of the axon, arborisation, and signalling to the dendritic spines, and the migration of developing neurons to their destinations, thus playing an essential role in cognitive functions and behaviours [11,14,30]. Numerous lines of evidence suggest the linkage of tubulin dysfunction with an increased risk of developing neuropsychiatric diseases such as major depressive disorder (MDD), bipolar disorder, and neurodevelopmental disorders, including ASD, schizophrenia, and epilepsy [29,67,68,69]. In the present study, in line with our previous results related to α/β-tubulin immunoreactivity obtained in the hippocampus and cortex in the aftermath of prenatal exposure to VPA [57], the level of cerebellar α/β-tubulin was also significantly reduced in VPA offspring. A significant decrease in the level of cerebellar α/β-tubulin may lead to damage to MT structure and could have deleterious effects on the cytoskeletal network. The loss of the α/β-tubulin may also be responsible for the morphological abnormalities found in nerve cells in the cerebellum of adolescent offspring exposed prenatally to VPA that are indicative of neuronal chromatolysis. Chromatolysis is associated with an adversely affected cytoskeletal network that frames the neuronal cytosol and supports the nucleus. This loss of nuclear suspension is responsible for the nucleus losing its central position and becoming eccentric, lying adjacent to the cell membrane [70,71]. Consistent with our results, the study by Sandhya et al. demonstrated that early prenatal or postnatal exposure to VPA led to histoarchitectural changes in the cerebellar neurons, accompanied by chromatolysis and neuronal degeneration [72]. It is also worth noting that chromatolysis is often associated with degranulation, disaggregation of polyribosomes, and degradation of monoribosomes into dust-like particles. This means that chromatolysis can be described as the disruption of the protein synthesis infrastructure [71]. Thus, VPA-induced fragmentation of Nissl substance and the probably subsequent progressive destruction of protein synthesis machinery may explain declines in protein levels observed in our study, i.e., α/β-tubulin, Tau, MAP1B, MAP2A/B, MAP2C/D, and αII-spectrin.

The polymerisation, stabilisation, arrangement, and function of MTs can be modulated by interactions with a series of MAP proteins [38]. Probably the most studied MAP is the MAP-Tau protein. Abnormal Tau hyperphosphorylation destabilise MTs, perturb MT function and axonal transport, contributing to synaptic miscommunication and neuronal degeneration [38]. Among more than 80 potential phosphorylation sites, phosphorylation at (Ser396/404) seems to play a pivotal role in Tau function and, in particular, depolymerises and destabilises MTs [43,44,73,74,75,76]. Several neurodegenerative diseases such as Alzheimer’s disease (AD), progressive supranuclear palsy, frontotemporal dementia, as well as various types of parkinsonism and synucleinopathies, are characterised by the occurrence of insoluble, highly phosphorylated Tau at (Ser396) [76,77,78,79]. In addition to (Ser396), (Ser199/202) and (Ser416) are crucial phosphorylation sites of Tau, which have been implicated in Tau pathology [80,81]. Moreover, in vitro studies show that (Ser199/202) is one of the critical phosphorylation sites that convert Tau to an inhibitory molecule that sequesters normal MAP from MTs [46,82].

Novel and more recent studies suggest an enabling role of Tau protein in the pathogenesis of ASD. The revealed changes in Tau protein expression in autistic patients [20,83,84], in autistic-like rodents [24,85], and in our previous studies [57,86], provide evidence suggesting the importance of Tau pathology in the aetiology of ASD and the formation of autistic-like behaviour. Our current study is the first in which we revealed a significant decrease in the level of Tau protein as well as its excessive hyperphosphorylation at (Ser396) together with a decrease in phosphorylation at (Ser416) in the cerebellum of adolescent offspring, as a result of exposure in utero to VPA. Abnormal phosphorylation and dysfunction of Tau, the manager of the neuronal cytoskeleton, may contribute to MT destabilisation, synaptic terminal dysfunction, and, consequently, cognitive and behavioural abnormalities [46] observed in our animals [55]. In contrast, Baron-Mendoza et al. showed a lack of changes in the content of p-Tau(Ser396) in the cerebellum of the autistic-like mouse strain C58/J [24]. We suggest that alterations in the Tau phosphorylation status are dependent on the cause and form of autism. It is likely that the discrepancy in the results may be due to differences in the experimental model of ASD. A genetic modification inducing the autistic-like behaviour in mice C58/J strain may evoke a different effect on Tau protein phosphorylation than the environmentally triggered rat model of ASD, which is based on prenatal exposure to VPA. Moreover, a model of environmental influence on brain function is highly pleiotropic, unlike various genetic models, and may better approximate the nature of ASD, which is extremely pleiotropic itself, involving a whole sequence of environmental and genetic ‘hits’ that disturb the homeostasis of the developing brain.

The excessive phosphorylation of Tau may be caused by conformational VPA-induced change(s) in Tau, which makes this protein more susceptible to phosphorylation [87]. Tau is a substrate for several protein kinases. Among them, GSK-3β, CDK5, ERK1/2, AMPK, and p70S6 kinase have been shown to phosphorylate Tau at several of the same sites as the abnormally hyperphosphorylated Tau seen in AD [87,88,89,90,91,92,93,94]. In our study, prenatal exposure to VPA induced significant activation of CDK5, AMPK, and p70S6 kinase without changes in GSK-3β and ERK1/2 activity. All these data suggest the potential, significant role of both CDK5 and AMPK, as well as p70S6 kinase, in VPA-evoked Tau hyperphosphorylation in the cerebellum of VPA rat offspring.

CDK5 is a serine-threonine kinase that is directly responsible for Tau phosphorylation [95]. Induced by stress conditions, Ca^2+^- and calpain-dependent overexpression of p25 protein and consequently prolonged overactivation of the CDK5/p25 complex produces hyperphosphorylation of Tau in the brain, observed in neurodegenerative diseases [96], during chronic stress [97], after perinatal exposure to Pb [98], and also after embryological exposure to VPA in hippocampal and cortical neurons [57]. In the current study, we indicated a significant rise in the level of p25 protein, suggesting that calpain-dependent cleavage of p35 has occurred, leading to CDK5 overactivation as a result of prenatal exposure to VPA. These changes were also accompanied by an increased ratio of SBDP protein to αII-spectrin (SBDP/αII-spectrin). SBDP protein is generated in the aftermath of the calpain-catalysed breakdown of αII-spectrin [99]. Altogether, our results demonstrate the calpain-dependent activation of CDK5 in the cerebellum following prenatal administration of VPA and suggest that CDK5 may be involved in VPA-evoked hyperphosphorylation of Tau in this brain structure. Additionally, it is noteworthy that αII-spectrin is one of the proteins essential for the cytoskeletal network, the depletion of which may lead to cytoskeletal damage, just like in the prolonged activation of calpains. Abundant evidence suggests that prolonged up-stimulation of calpain may lead to cytoskeleton disruption and neuronal death [100,101].

The next kinase, which has been implicated in the pathogenesis of ASD and could phosphorylate Tau, is AMP-activated protein kinase (AMPK). AMPK phosphorylates Tau at several epitopes, including (Ser396), whereas (Ser199) and (Ser202) were not found to be direct AMPK substrates [89]. In the brain of AD patients, deregulated AMPK co-localises with phosphorylated Tau in pre-tangle and tangle-bearing neurons [89]. Additionally, the neuronal cytoskeletal actin, α/β-tubulin, and neurofilament have all been identified as potential AMPK substrates [102]. The molecular mechanism associated with the activation of AMPK involves an AMP-binding-dependent conformational change in the α-subunit activation loop of the kinase, which enables the phosphorylation of the activating residue (Thr172) [103]. Here we have shown a significant increase in p-AMPK(Thr172) level in the cerebellum of VPA rats without changes in the total levels of AMPK, thus indicating activation of this enzyme. Similarly, up-regulation of AMPK was demonstrated in the prefrontal cortex and cerebellum of the VPA-induced rat model of autism [104]. Moreover, using primary mouse and human hepatocytes, it has been established that VPA is a novel activator of AMPK [105]. Increased phosphorylation of AMPK at (Thr172), consistent with aberrant AMPK activation, was also observed in the BTBR mouse model of ASD [106]. It was also shown that overactivation of AMPK during a critical phase of neuronal development inhibits neuronal polarisation and disruption of PI3K transport in growing axons [107]. Moreover, activation of energetic stress signals and stimulation of AMPK cause inhibitory effects on neuronal development at multiple stages, including axon outgrowth, dendrite growth, and arborisation [108].

Compelling evidence indicates that the phosphorylation of MAP-Tau may also be regulated through the 70 kDa p70 ribosomal protein S6 kinase (p70S6K, also known as S6K) [91,92,94,109]. The study by An et al. revealed a significant correlation between the up-regulation of p70S6K activity and the progressive sequence of neurofibrillary pathology according to Braak’s criteria [92]. The accumulation of paired helical filaments (PHF)-Tau and up-regulation of Tau translation associated with phosphorylation/activation of p70S6K was increased in the brain of AD individuals. Additionally, the study by Pei et al. showed a close relationship between phospho-p70S6K(Thr421/Ser424) and phospho-Tau(Ser396/404) [91]. In the current study, we indicated a significant increase in the level of phospho-p70S6K(Ser371) and activation of the mTOR complex 1 (mTORC1), which points to the stimulation of this kinase after exposure to VPA during embryonic development. Thus, we suggest that p70S6K may be involved in the VPA-induced hyperphosphorylation of Tau in the cerebellum of rat offspring.

Deregulation of mTOR is associated with several neurological conditions and has been suggested in ASD pathology [110]. The activation of the mTOR signalling cascade may enhance Tau pathology by elevating the levels of Tau and its phosphorylation [94,109]. However, in the current study, we have noted a distinct depletion (about 38% vs. control) in the level of total Tau following treatment with VPA. These changes in the Tau protein level, as well as in the level of other analysed MAP proteins, might be a consequence of the decrease in protein translation as a result of chromatolysis and disruption of parts of the protein synthesis infrastructure despite the activation of mTORC1. Moreover, the observed loss of cerebellar Tau and other MAP proteins (α/β-tubulin, MAP1B, and MAP2) may also be a result of the AMPK-stimulated autophagy. A growing body of evidence indicates the role of autophagy in the clearing and reduction of Tau [111]. Additionally, the decrease in Tau levels might be a consequence of the alterations in its cleavage by proteases [112]. Hyperphosphorylation of Tau may alter its degradation (by both the proteasome and autophagy systems) and its truncation by proteases [112], which can affect total Tau levels. Although abnormal hyperphosphorylation and Tau aggregation make it resistant to proteolysis by calcium-activated neutral proteases [46], there are also data suggesting that calpain-mediated Tau cleavage precedes Tau phosphorylation in Aβ-treated hippocampal neurons, suggesting that it might be an early event in the pathological process shared by multiple tauopathies. Up-regulation of calpain activity and production of toxic fragments of Tau precedes Tau phosphorylation and the loss of synaptic proteins in the AD brain [113]. Perhaps the same relationship is possible in the VPA-exposed cerebellum, which could also explain the decrease in Tau levels appearing together with calpain up-regulation in our experimental conditions. However, we cannot exclude the involvement of other proteases in the VPA-induced loss of Tau protein in the cerebellum.

The function and organisation of MT, especially neuronal MT, depend on different MAPs [38], which contribute to normal cytoskeleton organisation and dendritic arborisation, which are crucial for the function and formation of neural networks [29]. Increasing evidence suggests that the levels of various MAPs are altered in neurodevelopmental disorders [11]. However, systematic investigation of the MAP proteins involved in the mediation of autism-like behaviours is lacking. To the best of our knowledge, this is the first study reporting alterations of MAPs in the cerebellum of VPA offspring. Our study revealed that exposure to VPA during embryonic development evoked a large depletion in cerebellar MAP1B, MAP2A/B as well as MAP2C/D levels.

MAP1B is a known modulator of MT dynamics and stability. MAP1B is essential for the development and function of the nervous system [48]. It is expressed in axons, dendrites, and growth cones and is regarded as the dominant protein in the formation and development of axons and dendrites [11,33,49]. MAP1B-deficient mice have an impairment in brain development [49,114]. In turn, MAP1B knockout neurons display delayed synaptic vesicle (SV) fusion events, a reduced density of synaptic contacts, decreased SV and dense core vesicle (DCV) density at presynaptic terminals, as well as an increased proportion of excitatory immature symmetrical synaptic contacts [50].

Depletion of MAP2 expression and laminar cytoarchitectonic changes were also revealed in adult autistic individuals in a study by Mukaetova-Ladinska et al. [13]. MAP2 interacts with MTs via the α/β-tubulin-binding domain and modifies the structure and stability of MTs, neuronal morphogenesis, cytoskeletal dynamics, and organelle trafficking in axons and dendrites [39]. MAP2 also cross-links actin filaments and interacts with the neurofilaments of the cross-bridges between MT and neurofilaments [115]. Phosphorylation of MAP2 modulates its binding affinity to MT. Several kinases, including CDK5, GSK-3β, PKA, PKC, JNK1, and ERK phosphorylate Ser/Thr-Pro motifs of MAP2 [29,39], whereas phosphatase-1, -2A, -2B, and -2C are responsible for its dephosphorylation [29]. Importantly, in the context of the hyperglutamatergic theory of autism [116], MAP2 phosphorylation is also regulated by synaptic activity (glutamate acting through NMDA receptor induces rapid dephosphorylation of MAP2 at (Ser136)) [117]. In our study, exposure to VPA induced a significant decrease in the level of both p-MAP2A/B and p-MAP2C/D phosphorylated at (Ser136). Therefore, it is plausible that a large loss in p-MAP2(Ser136) in the VPA-exposed cerebellum might be due to the overstimulation of glutamatergic signalling. Ziemińska et al. have demonstrated disturbances of neuroactive amino acid homeostasis (changes in Glu/Gln/GABA concentrations) in the rat brain after exposure to VPA consistent with the hypothesis suggesting the role of the glutamatergic disturbances in ASD pathology [118]. Additionally, it is tempting to speculate that an observed drastic loss in total MAP2 content may provide fewer substrates for phosphorylation under conditions of appropriate MAP2 kinases activation. Moreover, MAP2 is rich in polypeptide sequences enriched in proline, glutamic acid, serine, and threonine-(PEST) sequences, which are putative signals for rapid proteolytic degradation [119,120]. Studies show that activated calpain degrades MAP2, which inhibits MT assembly. Calpain-induced MAP2 degradation decreases the association of this protein with α/β-tubulin, leading to disruption of MT structure [39,121]. Perhaps the same phenomenon takes place in our experimental conditions, thus explaining the deficiencies in the level of not only MAP2 but also MAP1B.

Here, we also revealed a distinct reduction in the level of αII-spectrin (280kDa), an actin-crosslinking and molecular scaffold protein, which is one of the major cytoskeletal components of the cortical membrane of presynaptic terminals and axons [122]. Some studies implicate αII-spectrin in critical aspects of dendritic and axonal development and synaptogenesis [123]. The loss of αII-spectrin observed in our study, which is probably a result of calpain-induced cleavage of this protein and SBDP generation, may lead to cytoskeletal damage. Disruption of the αII-spectrin scaffold in neuronal or neurosensory cells may be a common final pathway of neurodegeneration or malfunction. Mutations in *SPTAN1* that encodes αII-spectrin cause severe and usually lethal neurodevelopmental defects. Disturbances in this and other spectrins are implicated in diverse degenerative and psychiatric conditions [124]. Mice lacking αII-spectrin experience embryonic lethality due to nervous system malformations [125]. Mice without αII-spectrin in the peripheral nervous system exhibit impaired neuronal excitability and axonal defects [126]. Human mutations in *SPTAN1* associated with West Syndrome are characterised by intellectual disability, agenesis of the corpus callosum, and hypomyelination [123,127]. Other *SPTAN1* neurological disorders include juvenile-onset hereditary motor neuropathy and hereditary spastic paraplegia [128,129,130]. However, there have been no studies on αII-spectrin defects in ASD so far. The deficiencies in the level of αII-spectrin, in addition to the loss of α/β-tubulin, Tau, MAP1B, MAP2A/B, and MAP2C/D, as well as Tau excessive phosphorylation together with chromatolytic phenotype in cerebellar neurons clearly indicate destabilisation and thus dysfunction of the MT cytoskeleton network in this brain structure.

The strength of our study is demonstrating for the first time that in the cerebellum of autistic-like rats, there is a significant decrease in the level of a wide spectrum of cytoskeletal proteins along with excessive MAP-Tau phosphorylation and key Tau-kinases activation. However, our study also has some limitations. The most serious limitation is that we did not answer the question of how these cytoskeletal protein alterations impact cerebellar functions. Cytoskeletal proteins such as MTs are closely associated with cellular processes such as synaptic vesicles’ transport, recycling, and endocytosis. Therefore, when MT homeostasis is deregulated, there are MT-dependent alterations in synaptic transmission and plasticity. Future research should focus on an analysis of the physiological and functional neuronal changes induced by the observed cytoskeletal instability. It would also be worth investigating whether specific cytoskeletal modulators could improve the behaviour of autistic-like animals to select new clinical intervention targets.

## 5. Conclusions

Our study has, for the first time, revealed that maternal exposure to VPA contributes to a significant reduction in the level of a wide spectrum of microtubule-associated proteins, including α/β-tubulin, Tau, MAP1B, MAP2A/B, MAP2C/D and αII-spectrin in the cerebellum of adolescent rat offspring. Moreover, prenatal exposure to VPA induced Tau protein hyperphosphorylation (a significant increase in the level of pTau(Ser396)), accompanied by up-stimulation of CDK5, AMPK, and p70S6K. All these kinases could be involved in Tau hyperphosphorylation in this brain structure. Activation of calpain has been proposed as one of the potential triggers of a molecular cascade leading to the loss of MT and cytoskeletal proteins (Figure 12). In addition, it seems that VPA-induced chromatolysis and, thus, subsequent progressive destruction of protein synthesis machinery could play a pivotal role in the depletion of microtubule-associated and cytoskeleton proteins. All the observed abnormalities in MAPs, along with the histopathological changes seen in the cell body of neurons (Purkinje cells), provide evidence that prenatal exposure to VPA may induce destabilisation and, thus, dysfunction of the MT cytoskeleton network. In consequence, the disturbed MT cytoskeleton may lead to synaptic pathology, impair neuronal communication, and predispose the brain to the development of autistic-like behaviours and neurodevelopmental deficits.

## Figures and Tables

**Figure 1 biomedicines-10-03031-f001:**
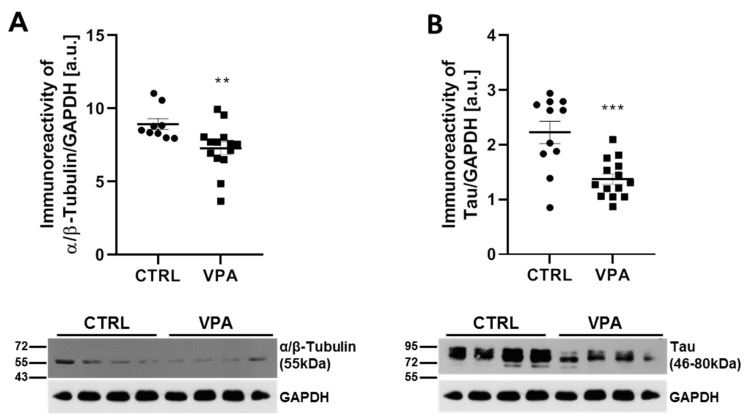
The effect of exposure to VPA during embryonic development on the level of α/β-tubulin and Tau protein in the cerebellum of adolescent rat offspring. Immunoreactivity of α/β-tubulin and Tau protein was analysed by Western blot technique. Densitometric analysis and representative pictures of α/β-tubulin (**A**) and Tau (**B**) are presented. Results were normalised to GAPDH levels. Data represent the mean values ± SEM from *n* = (9–14) independent experiments (number of separate animals from three different litters) for α/β-tubulin and *n* = (11–14) for total Tau. ** *p* < 0.01, *** *p* < 0.001, vs. control.

**Figure 2 biomedicines-10-03031-f002:**
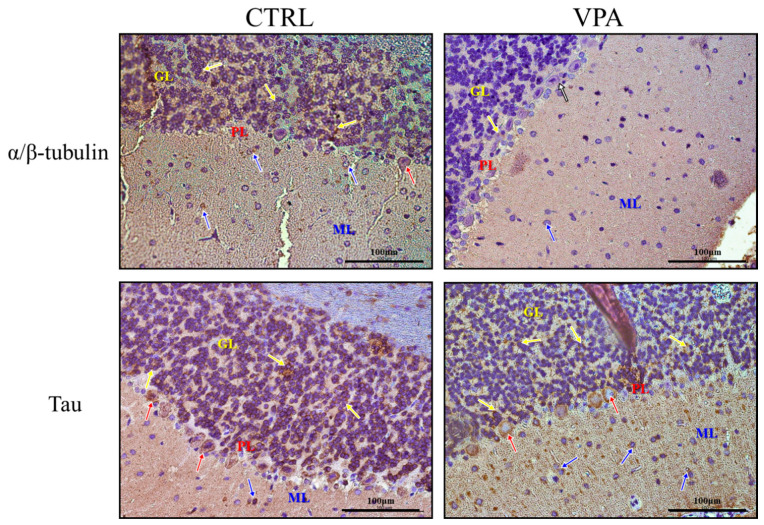
The effect of prenatal exposure to VPA on the immunoreactivity of α/β-tubulin and Tau protein in the cerebellum of adolescent rat offspring. Representative photomicrographs show the immunoexpression of α/β-tubulin and Tau protein in the cerebellum of control rats and VPA-treated rats. IHC reaction. Scale bar: 100 μm (objective magnification ×40). Yellow arrows—neurons of the granular layer (GL), red arrows—Purkinje cells (PL, the Purkinje cell layer), blue arrows—neurons of the molecular layer (ML), and white arrows—neurons with lack of immunoreactivity. Representative pictures from *n* = 6 (CTRL) and *n* = 6 (VPA) independent experiments (number of separate animals from three different litters) are presented.

**Figure 3 biomedicines-10-03031-f003:**
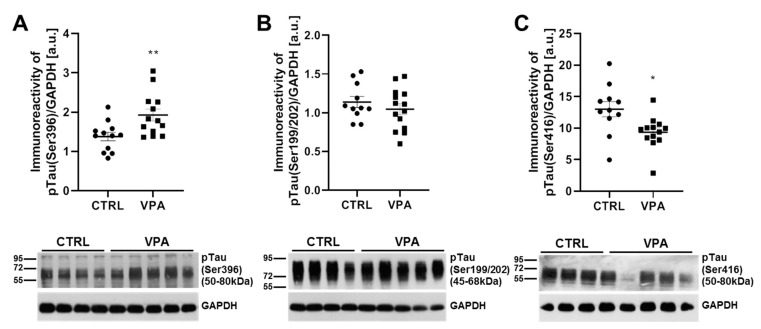
The effect of exposure to VPA during embryonic development on the phosphorylation state of Tau in the cerebellum of adolescent rat offspring. Immunoreactivity of pTau(Ser396), pTau(Ser199/202), and pTau(Ser416) protein was analysed by Western blot technique. Densitometric analysis and representative pictures of pTau(Ser396) (**A**), pTau(Ser199/202) (**B**), and pTau(Ser416) (**C**) in the cerebellum are presented. Results were normalised to GAPDH levels. Data represent the mean values ± SEM from *n* = (11–14) independent experiments (number of separate animals from three different litters). * *p* < 0.05, ** *p* < 0.01, vs. control.

**Figure 4 biomedicines-10-03031-f004:**
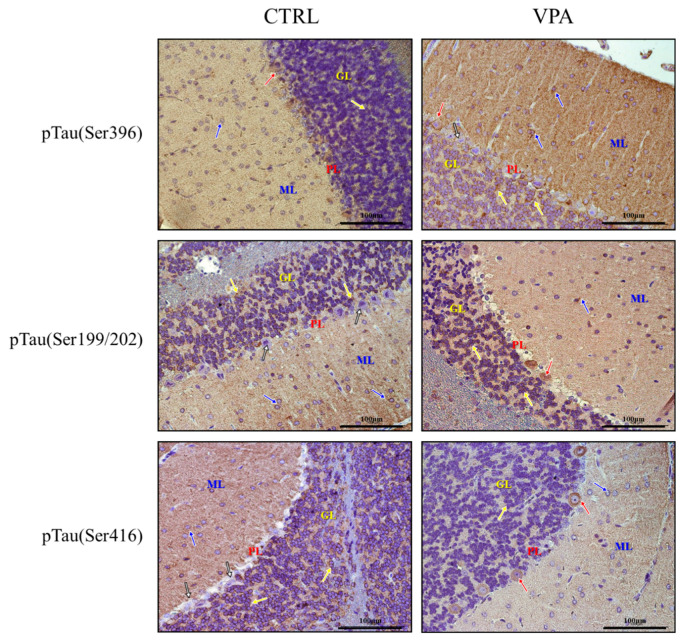
The effect of prenatal exposure to VPA on the immunoreactivity of Tau phosphorylated at (Ser396), (Ser199/202), and (Ser416) in the cerebellum of adolescent rat offspring. Representative photomicrographs show the immunoexpression of pTau(Ser396), (Ser199/202), and (Ser416) in the cerebellum of control rats and VPA-treated rats. IHC reaction. Scale bar: 100 μm (objective magnification ×40). Yellow arrows—neurons of the granular layer (GL), red arrows—Purkinje cells (PL, Purkinje cell layer), blue arrows—neurons of the molecular layer (ML), and white arrows—neurons with lack of immunoreactivity. Representative pictures from *n* = 6 (CTRL) and *n* = 6 (VPA) independent experiments (number of separate animals from three different litters) are presented.

**Figure 5 biomedicines-10-03031-f005:**
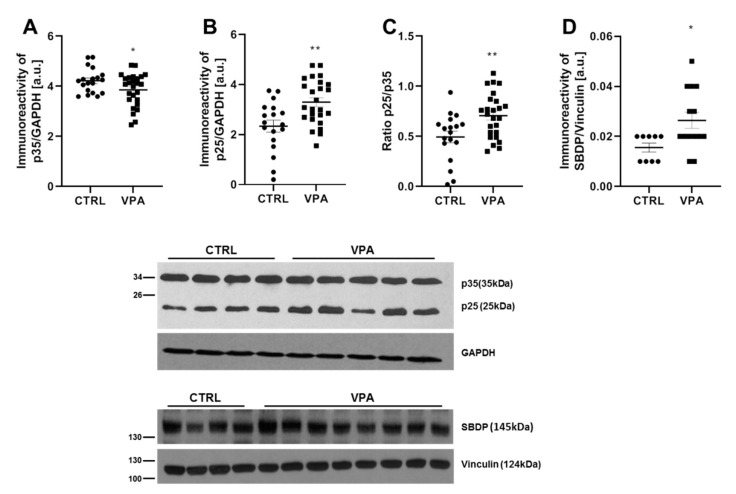
The effect of prenatal exposure to VPA on the calpain-dependent activation of CDK5 kinase in the cerebellum of adolescent rat offspring. Immunoreactivity of p35, its degradation product p25, and calpain-catalysed 145 kDa αII-spectrin breakdown product (SBDP) protein were determined using Western blot analysis. Densitometric analysis and representative pictures of p35 (**A**), p25 (**B**), and SBDP (**D**) in the cerebellum are shown. Results were normalised to GAPDH or vinculin levels. Additionally, the ratio of p25/p35 (**C**) was measured. Data represent the means ± S.E.M. from *n* = (18–26) independent experiments (number of separate animals from three different litters) for p35 and p25 protein and *n* = (9–14) for SBDP. * *p* < 0.05, ** *p* < 0.01, vs. control.

**Figure 6 biomedicines-10-03031-f006:**
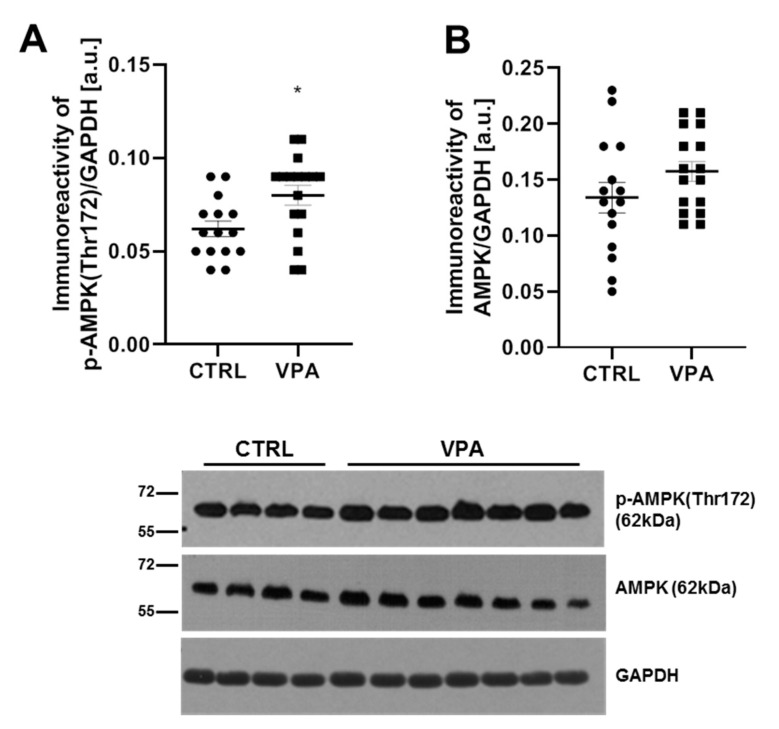
The effect of exposure to VPA during embryonic development on the AMPK in the cerebellum of adolescent rat offspring. Immunoreactivity of p-AMPK(Thr172) and AMPK were analysed by Western blot technique. Densitometric analysis and representative pictures of p-AMPK(Thr172) (**A**) and AMPK (**B**) in the cerebellum are presented. Results were normalised to GAPDH levels. Data represent the means ± S.E.M. from *n* = (15–17) independent experiments (number of separate animals from three different litters). * *p* < 0.05, vs. control.

**Figure 7 biomedicines-10-03031-f007:**
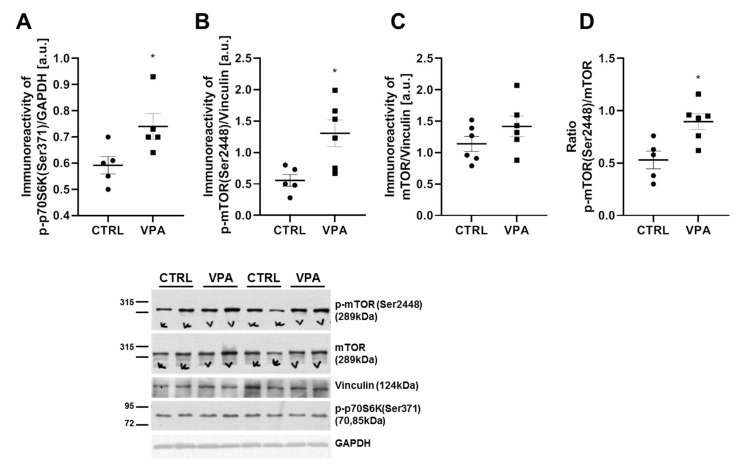
The effect of prenatal exposure to VPA on the p70S6K-mTOR downstream effectors in the cerebellum of adolescent rat offspring. Immunoreactivity of p-p70S6K(Ser371), p-mTOR(Ser2448), and total mTOR were determined using Western blot analysis. Representative blots and densitometric analysis of p70S6K phosphorylated at (Ser371) (**A**), p-mTOR(Ser2448) (**B**), mTOR (**C**), and ratio p-mTOR(Ser2448)/mTOR (**D**) are shown. Results were normalised to GAPDH or vinculin levels. Data represent the means ± S.E.M. from *n*= (5–6) independent experiments (number of separate animals from three different litters). * *p* < 0.05, vs. control.

**Figure 8 biomedicines-10-03031-f008:**
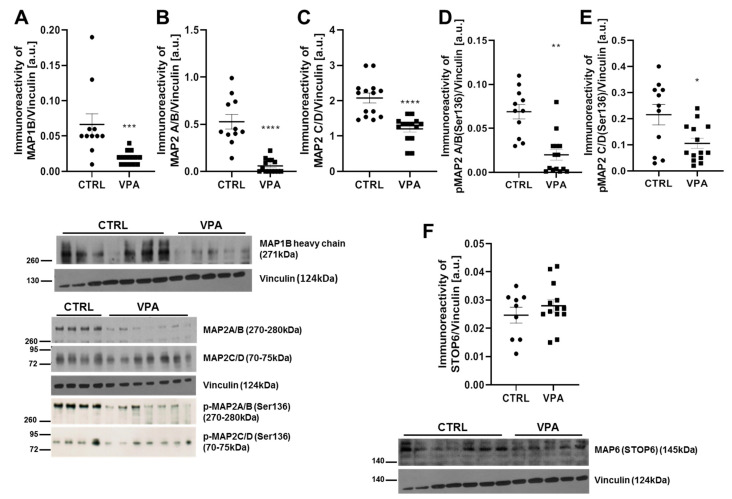
The effect of prenatal exposure to VPA on the MAP: MAP1B, MAP2A/B, MAP2C/D, p-MAP2A/B(Ser136), p-MAP2C/D(Ser136), and MAP6 (STOP) in the cerebellum of adolescent rat offspring. Immunoreactivity of MAP1B, MAP2A/B, MAP2C/D, phospho-MAP2A/B(Ser136), phospho-MAP2C/D(Ser136), and MAP6 (STOP) was monitored using Western blot analysis. Densitometric analysis and representative pictures of MAP1B (**A**), MAP2A/B (**B**), MAP2C/D (**C**), p-MAP2A/B(Ser136) (**D**), p-MAP2C/D(Ser136) (**E**), and MAP6 (STOP) (**F**), in the cerebellum are shown. Results were normalised to vinculin levels. Data represent the means ± S.E.M. from *n* = (9–14) independent experiments (number of separate animals from three different litters). * *p* < 0.05, ** *p* < 0.01, *** *p* < 0.001, **** *p* < 0.0001, vs. control.

**Figure 9 biomedicines-10-03031-f009:**
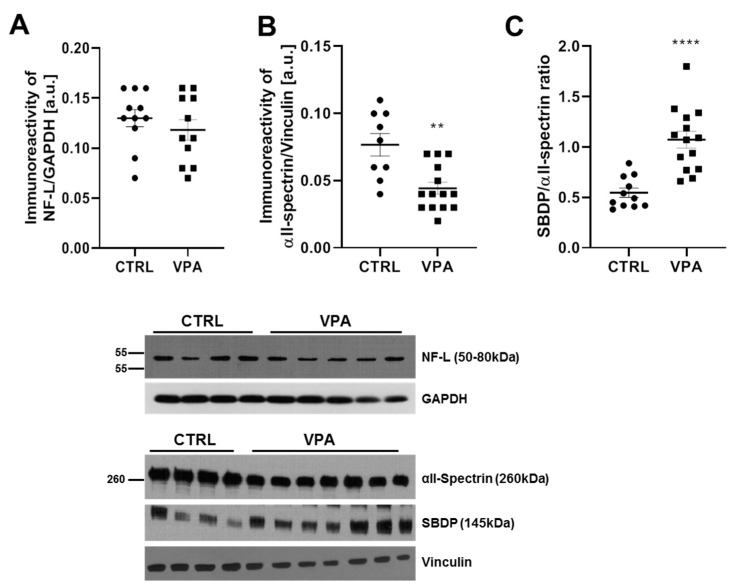
The effect of prenatal exposure to VPA on the level of cytoskeletal protein: NF-L and αII-spectrin in the cerebellum of adolescent rat offspring. Immunoreactivity of NF-L and αII-spectrin protein in control and VPA-exposed rats were monitored using Western blot analysis. Densitometric analysis and representative pictures of NF-L (**A**), αII-spectrin (**B**), and ratio SBDP/αII-spectrin (**C**) are shown. Results were normalised to GAPDH or vinculin levels. Data represent the mean values ± SEM from *n* = (9–14) independent experiments (number of separate animals from three different litters). ** *p* < 0.01, **** *p* < 0.0001, vs. control.

**Figure 10 biomedicines-10-03031-f010:**
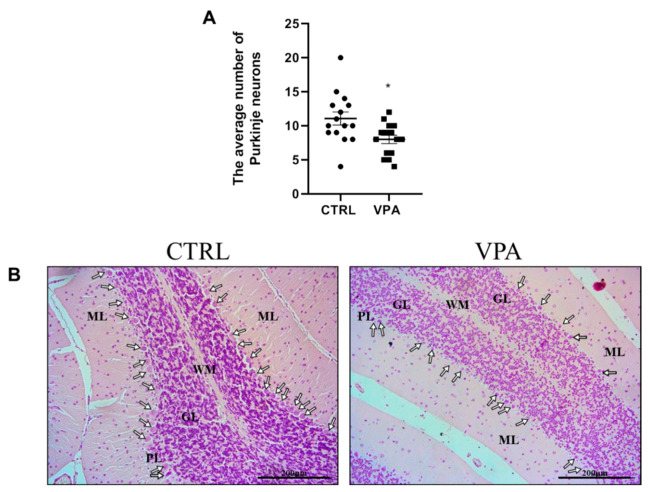
The effect of prenatal exposure to VPA on the number of Purkinje cells in the cerebellum of adolescent rat offspring. (**A**) Data represent the average number of Purkinje cells ± S.E.M from *n* = 5 control rats and *n* = 5 VPA-exposed rats. From each rat, three photomicrographs in the representative places of the cerebellum were taken, and all Purkinje cells visible in the field of view were counted and subjected to statistical analysis. * *p* < 0.05, vs. control. (**B**) The representative photomicrographs showing a large number of Purkinje cells (white arrows) in the cerebellum of a control rat compared to the reduced number of these cells in the VPA-exposed rat. ML—molecular layer, PL—Purkinje cells layer, GL—granular layer, WM—white matter. HE staining. Scale bar: 200 µm (objective magnification ×20).

**Figure 11 biomedicines-10-03031-f011:**
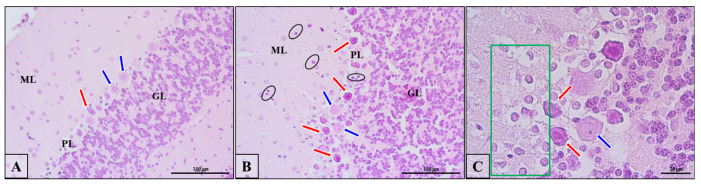
The effect of prenatal exposure to VPA on the histopathological changes in the neurons of the cerebellum of adolescent rat offspring. Representative photomicrographs show the histological structure of neurons in control (**A**) and VPA-exposed rats (**B**,**C**). H&E staining. Scale bar: 100 µm (**A**,**B**) and 20 µm (**C**); objective magnification ×40, ×100, respectively). Blue arrows—normal Purkinje cells; red arrows—neurons that show chromatolysis (acidophilic degradation of Nissl bodies and pyknotic nucleus); black circles/ellipsoid—mononuclear small, dark staining cells resembling lymphocytes; green frame—relaxation in neuropil.

**Figure 12 biomedicines-10-03031-f012:**
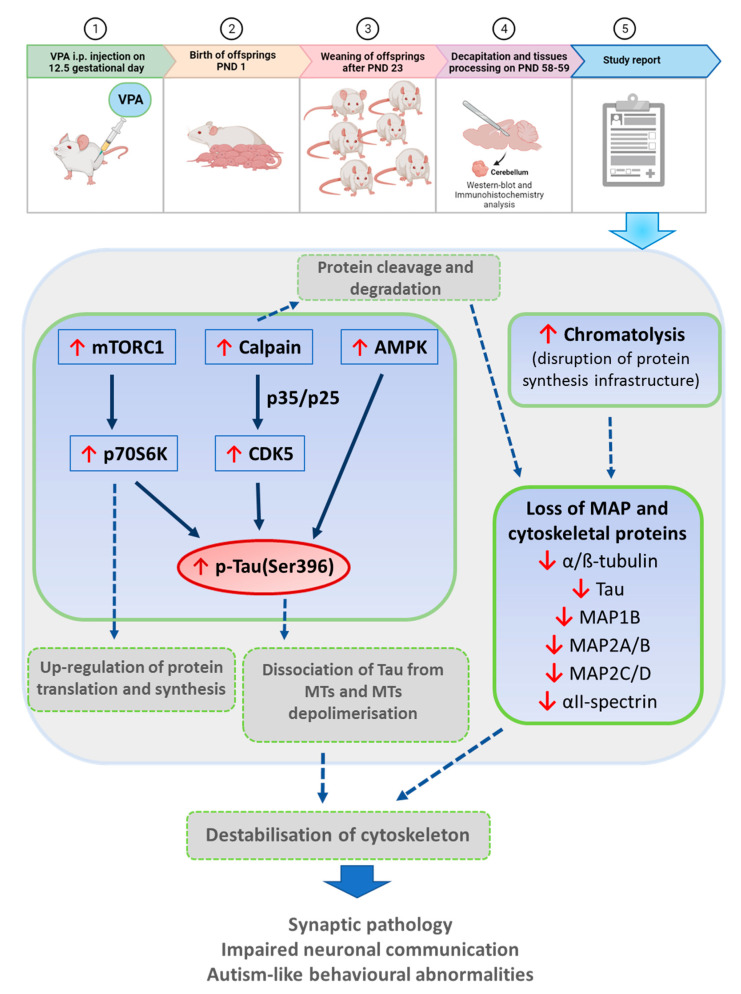
Schematic diagram showing the pathological changes in the cerebellum of adolescent male offspring after prenatal exposure to VPA. Maternal exposure to VPA (a single i.p. injection of VPA at 12.5 days of pregnancy) induced Tau protein hyperphosphorylation (a significant increase in the level of pTau(Ser396), accompanied by overstimulation of CDK5, AMPK, and p70S6K. All these kinases could be involved in VPA-induced Tau excessive phosphorylation in this brain structure. Abnormally phosphorylated Tau protein becomes dissociated from neuronal MTs, which results in α/β-tubulin depolymerisation and, consequently, destabilisation of MT cytoskeletal network. In addition, prenatal exposure to VPA led to significant decrease in the level of a wide spectrum of MAPs, including α/β-tubulin, Tau, MAP1B, MAP2A/B, MAP2C/D and αII-spectrin in the cerebellum of adolescent rat offspring, which could lead additionally to MT and cytoskeleton destabilisation. Activation of calpain and thus probably excessive protein cleavage and degradation, as well as chromatolysis and subsequent progressive destruction of protein synthesis machinery, have been proposed as one of the potential triggers of a molecular cascade leading to loss of MAP and other cytoskeletal proteins. All these abnormalities may lead to synaptic pathology, impairment axonal transport and neurotransmission, as well as predispose the brain to the development of autistic-like behaviours and neurodevelopmental deficits. This Figure was partially created using BioRender dynamic assets, adapted from “Mice Studies Workflow” by BioRender.com (2022). Retrieved from https://app.biorender.com/biorender-templates, access date: 27 April 2022.

## Data Availability

The raw data supporting the conclusions of this article will be made available by the authors on request, without undue reservation.

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
