# Peer review of "Alterations in Cerebellar Microtubule Cytoskeletal Network in a ValproicAcid-Induced Rat Model of Autism Spectrum Disorders"

_biomedicines, 2022, doi:10.3390/biomedicines10123031_

Round 1

Reviewer 1 Report

The study entitled “Alterations in cerebellar microtubule cytoskeletal network in a valproic acid-induced rat model of autism.” is an interesting piece of scientific achievement by the authors. However only one major concern remains unanswered.  During early brain development various deregulation like deregulation of the MT component is common and reported to be associated with various neurodevelopmental disorders. Autism and others are included in that, considering this how authors claim that the developed model is reflecting autism only. Autism is an multifactorial disease, considering this aspects, authors shall clarify appropriately before they claim induced rat model is autistic model, furthermore authors not able to reveal how these cytoskeletal proteins alterations impact cerebellar function.

Reviewer 2 Report

Gassowska-Dombrowolska et al. investigate changes of cytoskeleton-associated proteins in offspring of rats after treatment of the mothers with a high dose of valproic acid (VPA treatment during pregnancy is a toxic model for human autism). They focus on the cerebellum and find neuronal death (chromatolysis) by histological analysis. They used Western blots to quantify the expression of cytoskeletal elements and find reductions of tubulin and microtubule-associated proteins such as tau, MAP1B and spectrin. Tau protein was also hyperphosphorylated, and some tau kinases (cdk5, AMPK and p70S6K) were dysregulated. Overall, a fair amount of work is summarized in this manuscript.

All in all, this is careful work justified by recent propositions that a dysregulated cytoskeleton may play a role in autism and other neurological diseases. The manuscript is written in very good English and is free of grammatical mistakes. Methods and Results are clearly documented, the data are discussed in great detail and with numerous references to published literature. However, the Introduction and Discussion parts would be appropriate for a thesis or a review, but they are too long and verbose for an experimental paper. I therefore suggest to shorten and focus the manuscript to its essential points:

1.       In the Introduction, the authors cite a long list of review papers. I suggest to cite only two outstanding reviews per statement instead of citing several reviews for the same point, e.g. “[7-12]” in line 53 and “[13-26]” in line 58 or “[27-34]” in line 61. Thus, the enormous number of references (181 !) could be reduced.

2.       Some paragraphs in the Introduction are (more or less) repeated in the Discussion. These paragraphs can be deleted.  

3.       Table 1 can be shown as Supplemental Material.

4.       The Results part is okay but the Discussion is way too long. As one example, I suggest to delete some paragraphs that discuss taukinases that were not changed in the present study. Also, the text on MAPs is rather long. Fig. 12 nicely summarizes the contents of the paper but Table 2 seems superfluous.

Reviewer 3 Report

The authors created a rodent model of autism through a single prenatal administration of valproic acid (VPA) into pregnant rats, followed by cerebellar morphological studies of the offspring, focusing on the alterations of key cytoskeletal elements. They subsequently investigated the expression of α/β-tubulin and the major neuronal MT-associated proteins (MAP) such as MAP-Tau and  MAP1B, MAP2, MAP6 (STOP) along with actin-crosslinking αII-spectrin and neurofilament light polypeptide. The authors found that maternal exposure to VPA induces a significant de crease in the protein levels of α/β-tubulin, MAP-Tau, MAP1B, MAP2, and αII-spectrin. Moreover, excessive MAP-Tau phosphorylation at (Ser396) along with key Tau-kinases activation was indicated. Immunohistochemical staining showed chromatolysis in the cerebellum of autistic-like rats and loss of Purkinje cells. The paper has the potential to contribute to the existing scientific literature on the ASD, shedding light in one of the possible molecular mechanisms underpinning neuroplasticity alterations in the ASD brain. I only have a few comments to further improve the quality of the authors’ paper. I have outlined these issues below:

1. Although the authors adapted Schneider and Przewlocki 2004 paper to estabolish animal model of Autism, it would be better the authors can demonstrate physiological and/or behavioral changes in the offspring from their own laboratory to validate the model.

2. Was there any particular reason that the authors chose the adolescent rats (PND58-59) instead of PND30-50 in this study? 

3. It would be better the authors can demonstrate the dysregulation of microtubule-assoiciated proteins in the cerebellum of adolescent rat affect the cerebellum function and/or cerebellum-related behavior.

In the reviewer’s opinion, the above-mentioned issues need to be addressed by the authors.

Round 2

Reviewer 1 Report

Authors have incorporated sufficient changes both in the title and text. The revised manuscript shall be considered for publication. 

Reviewer 3 Report

The authors have revised their manuscript according to the reviewer's comment.  I have no further comments.